# Determination of topological edge quantum numbers of fractional quantum Hall phases by thermal conductance measurements

Saurabh Kumar Srivastav [1,9], Ravi Kumar[1,9], Christian Spånslätt [2], K. Watanabe [3], T. Taniguchi [3], Alexander D. Mirlin[4,5,6,7], Yuval Gefen[4,8] & Anindya Das [1] ✉

To determine the topological quantum numbers of fractional quantum Hall (FQH) states hosting counter-propagating (CP) downstream ($N_d$) and upstream ($N_u$) edge modes, it is pivotal to study quantized transport both in the presence and absence of edge mode equilibration. While reaching the non-equilibrated regime is challenging for charge transport, we target here the thermal Hall conductance $G_Q$, which is purely governed by edge quantum numbers $N_d$ and $N_u$. Our experimental setup is realized with a hexagonal boron nitride (hBN) encapsulated graphite gated single layer graphene device. For temperatures up to 35 mK, our measured $G_Q$ at $\nu = 2/3$ and 3/5 (with CP modes) match the quantized values of non-equilibrated regime $(N_d + N_u)\kappa_0 T$, where $\kappa_0 T$ is a quanta of $G_Q$. With increasing temperature, $G_Q$ decreases and eventually takes the value of the equilibrated regime $|N_d - N_u|\kappa_0 T$. By contrast, at $\nu = 1/3$ and 2/5 (without CP modes), $G_Q$ remains robustly quantized at $N_d \kappa_0 T$ independent of the temperature. Thus, measuring the quantized values of $G_Q$ in two regimes, we determine the edge quantum numbers, which opens a new route for finding the topological order of exotic non-Abelian FQH states.

In the quantum Hall (QH) regime, transport occurs in one-dimensional gapless edge modes, which reflect the topology of the bulk filling factor $\nu$. In integer QH (IQH) states and in a certain subclass of fractional QH (FQH) states, only downstream edge modes ($N_d$ of them) exist, whose chirality is dictated by the direction of the applied magnetic field[1,2]. At the same time, the edge structure of a majority of FQH states, including, in particular, the "hole-like" states ($1/2 < \nu < 1$), is more complicated. In addition to the downstream edge modes, the presence of upstream modes ($N_u$) leads to complex transport behavior[1–6]. In this situation, the measured values of the electrical conductance ($G_e$) depends on the extent of the charge equilibration between the counter-propagating downstream and upstream modes. For example, the $\nu = 2/3$ state hosts two counter-propagating modes: a downstream mode, $\nu = 1$, and an upstream $\nu = 1/3$ mode[3]. With full charge equilibration, the two-terminal conductance $G_e$ becomes[7–10] $2e^2/3h$; on the other hand, in the absence of charge equilibration, $G_e$ is equal to[8,10] $4e^2/3h$. The observation of a crossover from $4e^2/3h$ to $2e^2/3h$ is essential to establish the proposed edge structure. This crossover has indeed been observed in carefully engineered double-quantum-well structure, allowing control of the equilibration[11]. At the same time, a similar demonstration is lacking in experiments on a conventional edge (the boundary of a $\nu = 2/3$ FQH state), where $G_e$ is always found to be $2e^2/3h$.

[1]Department of Physics, Indian Institute of Science, Bangalore 560012, India. [2]Department of Microtechnology and Nanoscience (MC2), Chalmers University of Technology, S-412 96 Göteborg, Sweden. [3]National Institute of Material Science, 1-1 Namiki, Tsukuba 305-0044, Japan. [4]Institute for Quantum Materials and Technologies, Karlsruhe Institute of Technology, 76021 Karlsruhe, Germany. [5]Institut für Theorie der Kondensierten Materie, Karlsruhe Institute of Technology, 76128 Karlsruhe, Germany. [6]Petersburg Nuclear Physics Institute, 188300 St. Petersburg, Russia. [7]L. D. Landau Institute for Theoretical Physics RAS, 119334 Moscow, Russia. [8]Department of Condensed Matter Physics, Weizmann Institute of Science, Rehovot 76100, Israel. [9]These authors contributed equally: Saurabh Kumar Srivastav, Ravi Kumar. ✉e-mail: anindya@iisc.ac.in

The reason is that the small value of the charge equilibration length makes it difficult to access the nonequilibrated regime. A small deviation from $2e^2/3h$ indicating a beginning of the crossover towards $4e^2/3h$ was observed for the spin-unpolarized $v = 2/3$ FQH state[12].

Measurements of the thermal conductance have recently emerged as a powerful tool to detect the edge structure of FQH states[13–18]. Such measurements are highly useful for "counting" edge modes and can also detect charge neutral Majorana modes[16,19]. For IQH states and FQH states with only downstream modes, the quantized thermal conductance is given by $G_Q = N_d \kappa_0 T$, where $\kappa_0 = \pi^2 k_B^2/3h$, $k_B$ is the Boltzmann constant, $h$ is the Planck constant, and $T$ is the temperature[14]. A schematic illustration of the heat flow for such a state ($v = 1/3$ in this example) is depicted in Fig. 1a. On the other hand, for hole-like FQH states, the presence of upstream modes renders the value of $G_Q$ strongly dependent on the extent of thermal equilibration between CP modes. This leads to a crossover[8] of $G_Q$ from a nonequilibrated quantized value of $(N_d + N_u)\kappa_0 T$ to the asymptotic value of full equilibration $|N_d - N_u|\kappa_0 T$. While the fully equilibrated and nonequilibrated limiting cases of $G_Q$ have been reported in disparate GaAs/

AlGaAs based 2DEG devices[15,16,20], and in graphene only the nonequilibrated values have been observed[18], an in situ crossover of $G_Q$ from the nonequilibrated to the fully equilibrated limit in a single device has remained unattainable.

The observation of crossover in $G_Q$ has remained one of the long-standing challenges on the path to reveal the detailed edge structure of the FQH states. For example, for $v = 2/3$, the by now "standard" model of the edge (based on the hierarchy construction[21,22]) suggests one downstream and one upstream mode[3,4,23]. At the same time, a 2/3 edge with two co-propagating downstream modes[1] would also correspond to a fully legitimate FQH edge from the point of view of general theory[23]. For the first case, $G_Q$ should exhibit a crossover with temperature. In the nonequilibrated regime, $L \ll \ell_{eq}^H$, where $L$ is the channel length and $\ell_{eq}^H$ is the thermal equilibration length, $G_Q = 2\kappa_0 T$, whereas in the equilibrated regime, $\ell_{eq}^H \ll L$, the $G_Q$ will exhibit asymptotic value $\approx 0\kappa_0 T$. Such a crossover of $G_Q$ is schematically depicted in Fig. 1b, c. On the other hand, for the second case, $G_Q$ will be independent of temperature with a value of $2\kappa_0 T$. Similarly, the $v = 3/5$ edge model corresponding to the hierarchy construction harbors one

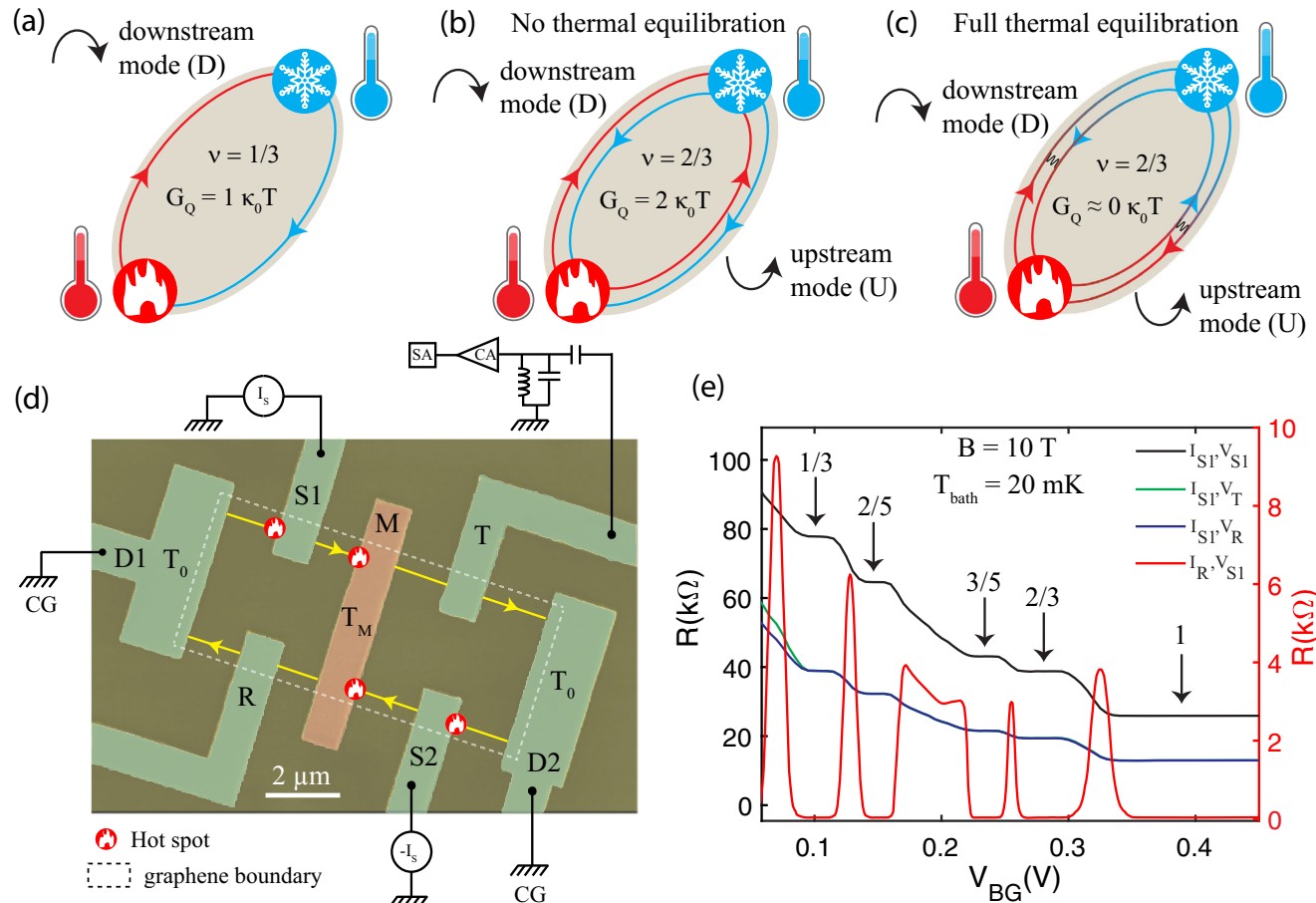

**Fig. 1 | Schematics of heat transport on QH edges, measurement setup, and QH response of device. a** Heat transport at the edge of $v = 1/3$ state along a single downstream mode. The chirality of the downstream mode is clockwise. **b** Heat transport at the edge of $v = 2/3$ state in nonequilibrated regime. Heat from the hot reservoir is carried away by both downstream and upstream modes. The chirality of upstream mode is anticlockwise. **c** Heat transport at the edge of $v = 2/3$ state in the equilibrated regime. The gradient of the color along the edges represents the qualitative temperature profile. In the long-length limit ($L \to \infty$), the heat carried away from the hot reservoir comes back to it via other edge modes, which leads to a vanishing thermal conductance. **d** False colored SEM micrograph of the device, shown with the measurement schematic. The graphene boundary is marked with a white dashed line. For illustrative purposes, the device is depicted with a $v = 1$ edge structure. For thermal conductance measurements, currents $I_S$ and $−I_S$ are fed

simultaneously at contacts $S1$ and $S2$. Due to the power dissipation near the central, floating contact, the electron temperature increases to $T_M$. The electrical and thermal conductances are measured respectively at low frequency (23 Hz) and high frequency (~740 kHz) with an LCR resonant circuit. **e** QH response: The black line is the resistance $R_{S1}$ ($V_{S1}/I_{S1}$) measured at source contact '$S1$' as a function of $V_{BG}$ at B = 10 T and temperature 20 mK. The blue line shows the measured resistance ($V_R/I_{S1}$) at the contact '$R$'. The green line shows the measured resistance ($V_T/I_{S1}$) at the contact '$T$'. The red curve shows the resistance $V_{S1}/I_R$ measured at the contact '$S1$', while the current is injected at the contact '$R$' and encodes the longitudinal resistance. Robust fractional plateaus at $\frac{3h}{e^2}$, $\frac{5h}{2e^2}$, $\frac{5h}{3e^2}$, and $\frac{3h}{2e^2}$ are clearly visible. The legend defines the current sources and voltage probes for each curve. The subscripts of $I$ and $V$ correspond to the current-fed contact and the voltage-probe contact, respectively.

downstream and two upstream modes[7,23], and as a result $G_Q$ will have a crossover from $3\kappa_0 T$ to $1\kappa_0 T$. However, there exist also alternative topologies (encoded by so-called $K$-matrices[23]) corresponding to $\nu = 3/5$. In particular, one can imagine a $\nu = 3/5$ edge with three co-propagating downstream modes[1], and in this scenario $G_Q$ would be independent of temperature with a value of $3\kappa_0 T$. Furthermore, the value of $G_Q$ can reveal the possible edge reconstruction of the QH states[24,25]. For example, for $\nu = 1/3$, the edge reconstruction by a pair of counter-propagating modes[26] would increase the number of modes from 1 to 3, implying a crossover of $G_Q$ from $3\kappa_0 T$ at low $T$ (none-quilibrated regime) to $1\kappa_0 T$ at higher $T$. Similarly, for $\nu = 2/3$, the edge reconstruction would increase the number of modes[27] from 2 to 4, which would result in $G_Q = 4\kappa_0 T$ at low temperature (nonequilibrated regime). Thus, the observation of crossover in $G_Q$ and its precise values can determine the exact topological number of the FQH edges. Achieving this goal would further help to settle the topological order of more complex non-Abelian even-denominator FQH states.

In this work, we report on thermal conductance measurements as a function of temperature ($T$) of electron-like ($\nu = 1/3$ and 2/5) and hole-like ($\nu = 2/3$ and 3/5) FQH states, realized in a hBN encapsulated graphite-gated high-mobility single layer graphene device. Our key findings are the following: (1) At the base temperature (lowest bath temperature $T_{bath}$, ~ 20 mK), $G_Q$ for 2/3 and 3/5 is found to be $2\kappa_0 T$ and $3\kappa_0 T$, respectively, and remain constant up to ~35 mK. (2) With further increase of temperature, $G_Q$ for 3/5 decreases, saturating at $1\kappa_0 T$ for $T \gtrsim 50$ mK. The similar crossover of $G_Q$ is observed for 2/3 too and $G_Q$ drops to a value ~$0.5\kappa_0 T$ at 60 mK, continuing to decrease toward zero. The observed values of $G_Q$ matches with the theoretical models for the hole-like FQH states with CP modes from the nonequilibrated limit of $(N_d + N_u)\kappa_0 T$ to the equilibrated limit of $|N_d - N_u|\kappa_0 T$. For $\nu = 2/3$, the heat transport in the equilibrated regime is of diffusive character, with the limiting value $|N_d - N_u|\kappa_0 T \approx 0$ that is approached in a power-law way as a function of temperature. (3) For 1/3 and 2/5 FQH states, $G_Q$ is found to be $1\kappa_0 T$ and $2\kappa_0 T$, respectively, independent of the electron temperature and matches with the expected $G_Q = N_d \kappa_0 T$ without CP modes. These observations further confirm that there is no edge reconstruction in our device.

## Results

### Device schematic and response

To measure the thermal conductance, we have used a graphite-gated graphene device, where the graphene is encapsulated between two hBN layers. The details of the device fabrication is described in Methods as well as in Supplementary Note 1. One of the important length scales of the device is the separation between the graphene and the screening graphite layer, which is ~25 nm and comparable to the magnetic length scale. It has been theoretically predicted[28] that for such cases edge reconstruction can be avoided (see Supplementary Note 12). We will below show that our measured $G_Q$ confirm the absence of edge reconstruction for our device. Similar to our previous work[17,18], our device consists of a small floating metallic reservoir, which is connected to graphene channel via one-dimensional edge contacts, as shown in Fig. 1d. To measure the electrical conductance, we used the standard lock-in technique whereas the thermal conductance measurement was performed with noise thermometry[15–18,29,30] (see Supplementary Fig. 2). In Fig. 1e, the black curve represents the measured resistance $R_{S1}$ ($V_{S1}/I_{S1}$) at the source contact ('S1') as a function of the graphite gate voltage ($V_{BG}$). Well developed plateaus appear at $\nu = \frac{1}{3}, \frac{2}{5}, \frac{3}{5}$, and $\frac{2}{3}$. The blue curve shows the measured resistance $R_R = V_R/I_{S1}$ along the reflected path (at contact 'R') from the floating contact. Similarly, the green curve shows the measured resistance $R_T = V_T/I_{S1}$ along the transmitted path (at contact 'T') from the floating contact. Measured resistances along the reflected and transmitted paths are identical, and exactly half of the resistance measured at the source contact, suggesting equal partitioning of

injected current to both the transmitted and reflected side (see Supplementary Note 4, and Supplementary Fig. 5). In fact, the equipartition of the current on both sides of the floating contact in Fig. 1e firmly establishes two important points: (i) it rules out the presence of any appreciable reflection coefficient at the interface of graphene and the floating contact (see Supplementary Fig. 6 for details), and (ii) the positions of the plateaus at the same gate voltage suggest the same electronic density on both sides of the floating contact. The red curve in Fig. 1e shows the resistance $R_{S1} = V_{S1}/I_R$ measured at contact 'S1', while the current is injected from the contact 'R'. This resistance in this configuration has the same properties as a longitudinal resistance: in the absence of bulk transport, the voltage $V_{S1}$ is determined by the equilibrium potential of the ground contact D1. The observation of the vanishing resistance plateaus further supports the formation of well developed FQH states. In Supplementary Fig. 7, we show plots analogous to Fig. 1e but measured at elevated temperatures within our working temperature range—without detectable changes either in resistance values or in equipartition of currents. It should be noted that the measured resistance values in Fig. 1e at the source, reflected and transmitted contacts suggest full charge equilibration in our device (see Supplementary Note 6 and Supplementary Table 2).

### Thermal conductance measurement

In contrast to our previous works[17,18], to measure the thermal conductance, we simultaneously inject the DC currents $I_S$ and $-I_S$ at two contacts $S1$ and $S2$, respectively. Both injected currents flow towards the floating reservoir. This is done in order to keep the potential of the floating contact to be the same as that of all drain contacts. In this configuration, the dissipated power at the floating reservoir due to Joule heating is given as $P = \frac{I_S^2}{\nu G_0}$ (see Supplementary Note 3). This power dissipation leads to increase of the electron temperature in the floating reservoir. The new steady state temperature $T_M$ is determined by the heat balance relation[15–18,29,31,32]

$$P = J_Q = J_Q^e(T_M, T_0) + J_Q^{e-ph}(T_M, T_0) = 0.5 N\kappa_0(T_M^2 - T_0^2) + J_Q^{e-ph}(T_M, T_0) \tag{1}$$

Here, $J_Q^e(T_M, T_0)$ is the electronic contribution of the heat current via $N$ chiral edge modes, and $J_Q^{e-ph}(T_M, T_0)$ is the heat loss via electron-phonon cooling, and $T_0$ is the electron temperature of the cold reservoirs. The temperature $T_M$ is obtained by measuring the excess thermal noise[15–18,29] along the outgoing edge channels using the Nyquist-Johnson relation

$$S_I = \nu k_B (T_M - T_0) G_0 \tag{2}$$

For our hBN encapsulated graphite-gated device[18], the electron-phonon contribution (second term in Eq. (1)) was found to be negligible for $T_{bath} < 100$ mK (see Supplementary Note 9 and Supplementary Fig. 12). From Eq. (1), one finds $N$, which yields the sought thermal conductance $G_Q = N\kappa_0 T$. It should be noted that Eq. (2) remains valid if there is a quasi-equilibrium state characterized by a hot Fermi distribution function with temperature $T_M$, which is satisfied for our device as the dwell time for the electrons in the metallic floating contact is longer than the electron–electron interaction time or thermalization time (see Supplementary Note 3). In Fig. 2, we show the detailed procedure to extract the quantized $G_Q$ at the bath temperature, $T_{bath}$ ~ 20 mK. Note that for each bath temperature, we experimentally determine the electron temperature, $T_0$ of the device and for our system $T_{bath} \approx T_0$ (see Supplementary Note 2, Supplementary Fig. 3, and Supplementary Table 1).

The measured excess thermal noise $S_I$ is plotted as a function of current $I_S$ for $\nu = 2/3$ and 3/5 in Fig. 2a, d, respectively. The resulting heating of the floating reservoir is made manifest by the increase in excess thermal noise with the application of the source current $I_S$. The

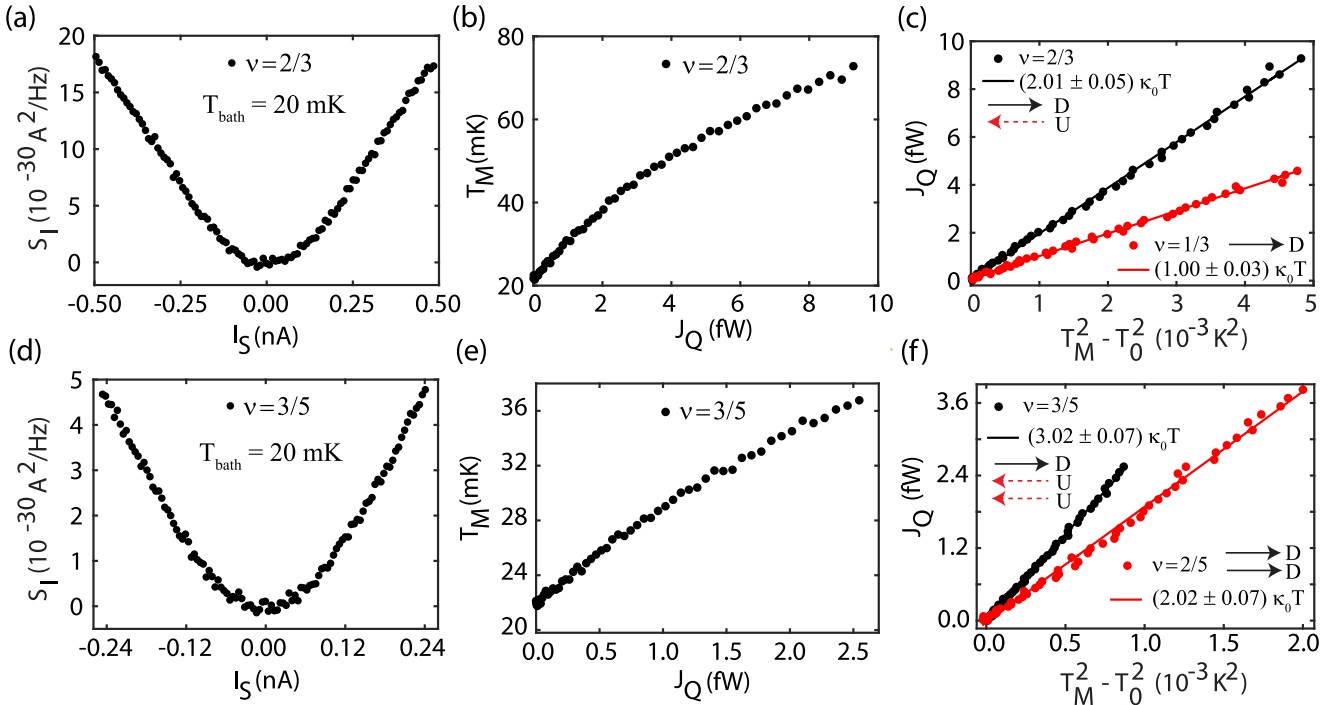

**Fig. 2 | Thermal conductance of fractional QH states. a** Excess thermal noise $S_I$ as a function of source current $I_S$ at $\nu = 2/3$. The DC currents $I_S$ and $-I_S$ were injected simultaneously at contacts $S1$ and $S2$, respectively, as shown in Fig. 1d. **b** The temperature $T_M$ of the floating contact as a function of the dissipated power $J_Q$ at $\nu = 2/3$. **c** $J_Q$ (solid circles) is plotted as a function of $T_M^2 - T_0^2$ at $\nu = 2/3$ (black) and 1/3 (red). Solid black and red lines are linear fits with $G_Q = 2.01\kappa_0 T$ and $1.00\kappa_0 T$ for $\nu = 2/3$ and 1/3, respectively. **d** Excess thermal noise $S_I$ as a function of source current $I_S$ at $\nu = 3/5$. **e** The temperature $T_M$ of the floating contact as a function of the dissipated power $J_Q$ at $\nu = 3/5$. **f** $J_Q$ (solid circles) is plotted as a function of $T_M^2 - T_0^2$ for $\nu = 3/5$ (black) and 2/5 (red). Solid black and red lines are linear fits with $G_Q = 3.02\kappa_0 T$ and $2.02\kappa_0 T$ for $\nu = 3/5$ and 2/5, respectively. The black and dashed red arrows depict the downstream and upstream modes, respectively, for each edge structure.

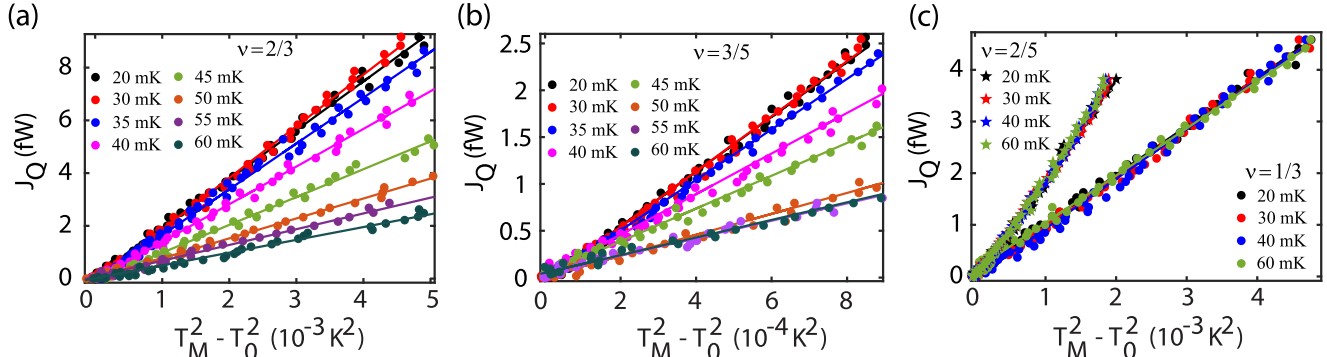

**Fig. 3 | Temperature dependence of thermal conductances. a, b** $J_Q$ (solid circles) is plotted as a function of $T_M^2 - T_0^2$ at $\nu = 2/3$ (**a**) and $\nu = 3/5$ (**b**) at several values of the bath temperature. Solid circles show the experimental data, while solid lines are linear fits to these experimental data points. Different colors correspond to different bath temperatures as shown in the legend. **c** $J_Q$ (solid circles) is plotted as a function of $T_M^2 - T_0^2$ for $\nu = 1/3$ (filled circles) and $\nu = 2/5$ (filled stars) at several values of the bath temperature. Different colors of the symbols correspond to different bath temperatures, (see legend). For all panels, the thermal conductance $G_Q$ at each temperature is extracted from the slope of the linear fit.

noise and current axes of Fig. 2a, d are converted to $T_M$ and $J_Q$, yielding Fig. 2b for $\nu = 2/3$ and Fig. 2e for $\nu = 3/5$, respectively. To extract $G_Q$, the heat current $J_Q$ is plotted as a function of $T_M^2 - T_0^2$ for $\nu = 1/3$ (red) and 2/3 (black) in Fig. 2c and for $\nu = 2/5$ (red) and 3/5 (black) in Fig. 2f. The solid circles represent the experimental data, while the solid lines are the linear fits with $G_Q = 1.00\kappa_0 T$ (red) and $2.01\kappa_0 T$ (black) for $\nu = 1/3$ and 2/3, respectively, in Fig. 2c and $G_Q = 2.02\kappa_0 T$ (red) and $3.02\kappa_0 T$ (black) for $\nu = 2/5$ and 3/5, respectively, in Fig. 2f. To further study the temperature dependence of the thermal conductance, $J_Q$ is plotted as a function of $T_M^2 - T_0^2$ at several values of the bath temperature for $\nu = 2/3$ in Fig. 3a and for 3/5 in Fig. 3b. An analogous plot is shown for $\nu = 1/3$

(solid circles) and 2/5 (solid stars) in Fig. 3c. The slopes of the linear fits to the data in these figures allow us to extract the values of $G_Q$. Whereas the data for the 2/3 and 3/5 states show an explicit dependence of $G_Q$ on bath temperature, the thermal conductance remains independent of the temperature for the 1/3 and 2/5 states, Fig. 3c. Note that the thermal conductance measurement was performed at the middle of each QH plateau.

In Fig. 4a, we plot the thermal conductance $G_Q$ (extracted from the slope of the linear fits to the data in Fig. 3 as a function of the bath temperature for $\nu = 1/3$ (red), 2/5 (blue), 2/3 (magenta), and 3/5 (black). As can be seen in Fig. 4a for $\nu = 1/3$ (red) and 2/5 (blue), the values $G_Q$

(a)
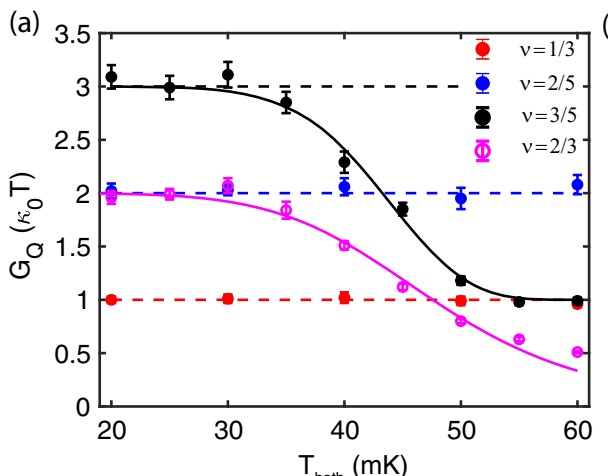

(b)
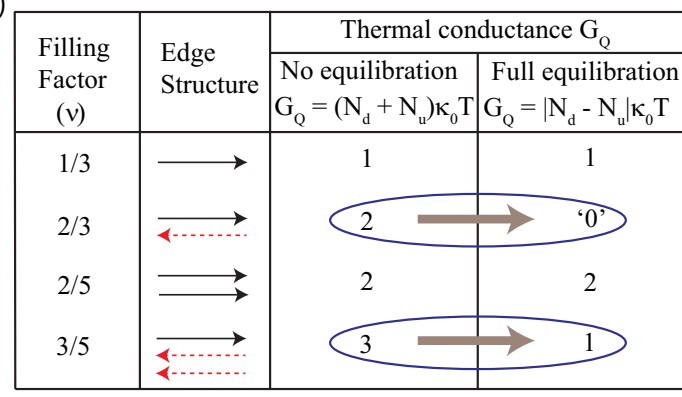

| Filling Factor ($\nu$) | Edge Structure | Thermal conductance $G_Q$ | |
|---|---|---|---|
| | | No equilibration $G_Q = (N_d + N_u)\kappa_0 T$ | Full equilibration $G_Q = \|N_d - N_u\|\kappa_0 T$ |
| 1/3 | | 1 | 1 |
| 2/3 | | 2 | '0' |
| 2/5 | | 2 | 2 |
| 3/5 | | 3 | 1 |

**Fig. 4 | Crossover from nonequilibrated to equilibrated heat transport.**
**a** Thermal conductance $G_Q$, as extracted from the slope of the linear fit in Fig. 3, plotted as a function of the bath temperature for $\nu = 1/3$ (red), 2/5 (blue), 3/5 (black), and 2/3 (magenta). The horizontal dashed lines correspond to quantized values of $G_Q$. The solid curves (black and magenta) are theoretical fits of the data that serve to extract out temperature scaling exponents (see Methods). Error bars correspond to the standard deviation associated with the slope of the linear fit shown in Fig. 3. **b** Edge structures of the studied FQH states. Solid black and dashed red arrows represent downstream and upstream modes, respectively. The two right-most columns show expected values of the thermal conductance $G_Q$ (in units of $\kappa_0 T$) in the two limiting regimes of the heat transport.

($1\kappa_0 T$ and $2\kappa_0 T$, respectively) remain independent of the bath temperature. The same behavior is found for integer QH states: $G_Q$ remains constant with temperature (see Supplementary Note 11 and Supplementary Fig. 13). On the other hand, for the hole-like 3/5 state, at the lowest bath temperature ($T_{bath} \sim 20$ mK), we observe $G_Q \sim 3\kappa_0 T$, which remain constant up to $T_{bath} \sim 35$ mK and with further increase of the temperature, the $G_Q$ decreases and saturates to ~$1\kappa_0 T$ for $T_{bath} \gtrsim 50$ mK. A similar crossover is observed also for the 2/3 state. For this state, at low temperatures, $G_Q \sim 2\kappa_0 T$ is observed. When the temperature increases beyond 35 mK, $G_Q$ starts decreasing and drops down to a value of ~$0.5\kappa_0 T$ at our largest temperature, $T_{bath} \sim 60$ mK.

To understand these results, we show the expected edge structures and their corresponding thermal conductance values for the studied FQH states in Fig. 4b. For the electron-like 1/3 and 2/5 states, there are only downstream modes with $N_d = 1$ and 2, respectively, and thus, the expected $G_Q$ should be $1\kappa_0 T$ and $2\kappa_0 T$, respectively, and will remain independent of the temperature. This is indeed seen for our experiment in Fig. 4a. This behavior is analogous to that for integer QH states (See Supplementary Note 11 and Supplementary Fig. 13), where all edge modes also propagate downstream. On the other hand, for the hole-like 3/5 state, the temperature dependence crossover of $G_Q$ from one quantum value to another one rules out any possibility of having only downstream modes. Furthermore, the measured values of $3\kappa_0 T$ and $1\kappa_0 T$, respectively, perfectly match with the nonequilibrated ($(N_d + N_u)\kappa_0 T$) and equilibrated ($|N_d - N_u|\kappa_0 T$) regimes of $G_Q$ with $N_d = 1$ and $N_u = 2$. Similarly, for 2/3, our observation rules out the theoretical model with only downstream modes, and support the crossover from the nonequilibrated regime of $G_Q$ to the equilibrated regime with $N_d = N_u = 1$. The equilibrated transport in this situation is diffusive in nature, so that $G_Q$ is expected to tend to zero relatively slowly (as ~$1/L$) in the long-length limit. Since our device channel length $L$ is limited to ~5 μm, we observe a finite value of ~$0.5\kappa_0 T$ at $T_{bath} \sim 60$ mK. Approaching substantially closer the asymptotic value of $0\kappa_0 T$ for 2/3 would be very interesting but it is not a simple task. For a given length $L$, this would require a further increase of temperature. However, we find that then the electron-phonon cooling starts to contribute significantly, spoiling the analysis (See Supplementary Note 9 and Supplementary Fig. 12).

Thus, measuring the quantized values of $G_Q$ at the two regimes helps to experimentally determine the topological edge quantum numbers. We note that, in our previous work[30], the noise measurement confirmed the presence of CP modes for hole-like FQH states in graphene. At the same time, the approach of ref. 30 was unable to detect exact topological edge quantum numbers. Furthermore, in the present study, the low-temperature values of $G_Q$ rules out any edge reconstruction in our device for all of the QH states studied. For example, for a 1/3 edge, one would observe a crossover from $3\kappa_0 T$ to $1\kappa_0 T$ in the presence of edge reconstruction. We find, however, $1\kappa_0 T$ down to lowest $T$, demonstrating that the edge reconstruction is not operative. Similarly, for a 2/3 edge, the edge reconstruction would increase the total number of edge modes from 2 to 4, and consequently would give rise to $4\kappa_0 T$ value in the low-$T$ limit instead of the observed $2\kappa_0 T$.

According to theoretical predictions, the crossover of $G_Q$ between the asymptotic limits of no thermal equilibration ($L \ll \ell_{eq}^H$) and perfect thermal equilibration ($L \gg \ell_{eq}^H$) is described by a function of the dimensionless ratio $L/\ell_{eq}^H$, with the thermal equilibration length scaling as a power of temperature, $\ell_{eq}^H \propto T^{-p}$. Explicit forms of the crossover functions for $\nu = 2/3$ and $\nu = 3/5$ states are given below in Methods. Our experimental data are well described by these forms as shown by the solid lines in Fig. 4a. At the same time, the values of the exponent $p$ that are obtained from the fits turn out to be unexpectedly large: $p = 6.3$ for $\nu = 2/3$ and $p = 9.3$ for $\nu = 3/5$, well above $p = 2$ expected in the vicinity of the strong-disorder fixed points[6-8]. This implies that the crossover $G_Q(T)$ is surprisingly sharp as a function of temperature. While this observation remains puzzling at this stage, several plausible equilibration mechanisms that might yield a large $p$ are discussed in the next section. It is worth noting that the asymptotic limit of $G_Q = 0\kappa_0 T$ in the equilibrated regime for 2/3 state is expected to be achieved (within our measurement accuracy) around $T_{bath} \sim 140$ mK [obtained by extending the fitted magenta curve in Fig. 4a], which is virtually impossible to experimentally measure due to strongly enhanced electron-phonon cooling as mentioned above.

## Discussion
In this section, we discuss a few additional points related to the expected theoretical regimes of equilibration, the accuracy of our measurement, the large temperature exponents of the thermal equilibration lengths, and future implications of our observations.

(1) The quantized value $G_Q = (N_d + N_u)\kappa_0 T$ of the thermal conductance in the nonequilibrated regime, $L \ll \ell_{eq}^H$, where $\ell_{eq}^H$ is the thermal equilibration length, strictly holds if there is no back-scattering of heat at interfaces with contacts. This is fulfilled under an additional condition $L \ll L_T$ where $L_T \sim T^{-1}$ is the thermal length. In the intermediate regime $L_T \ll L \ll \ell_{eq}^H$, a correction to this value is expected to emerge[8,20,33]. Thus, the nonequilibrated regime may, in fact, be expected to be split into two plateaus, which is, however, not observed in our experiment.

(2) The experimental determination of the thermal conductance follows the approach of several preceding works that use two implicit assumptions: (i) current fluctuations propagating from the central contact satisfy the thermal equilibrium distribution, implying the Johnson-Nyquist relation between the contact temperature and the noise; (ii) all power dissipated close to the central contact heats it. When all modes propagate downstream, both these assumptions strictly hold. However, for edges with CP modes, the situation may be somewhat more delicate and some deviations from the assumptions (i) and (ii) may emerge. This issue was discussed in ref. [20], where corrections to the procedure of extraction of $G_Q$ were obtained that slightly reduce the experimental value of $G_Q$. We do not include these corrections in the present work. First, they would not affect the identification of the asymptotic regimes. Second, the values of $G_Q$ that we find without including these corrections agree remarkably with the quantized values, both for the nonequilibrated regime (as was also found for bilayer graphene in ref. [18]) and in the equilibrated limit. It remains to see which features of our device favor this remarkable agreement. We would like to note that the precise determination of $G_Q$ depends on the accuracy of electron temperature and gain of the amplification chain, which are shown in details in Supplementary Note 2 and Supplementary Fig. 3.

(3) It was pointed out above that the temperature-driven crossover from nonequilibrated to equilibrated regime is remarkably sharp in our experiment, i.e., the parameter $p$ controlling the scaling of the equilibration length ($\ell_{eq}^H \propto T^{-p}$) is unusually large. Theoretically, the value of $p$ is controlled by irrelevant operators within the renormalization-group framework. Various mechanisms corresponding to such operators are known that may lead to large values of $p$ in correlated 1D systems. In particular, this may happen if the energy relaxation is dominated by complex (multiparticle) interchannel processes[7,23,34] or by nonlinearities of the quasiparticle and plasmon spectrum at the edge[35–41]. We leave a detailed investigation of this issue in the present context to future research.

(4) Observing a crossover of the thermal conductance between two asymptotic limits of the thermal equilibration is an important step toward pinpointing the topological order of more complex FQH states. Of particular interest is the $\nu = 5/2$ state, whose topological order is a subject of active debate. Specifically, the anti-Pfaffian state should demonstrate a crossover from 4.5 to $1.5\kappa_0 T$, whereas the PH-Pfaffian state should demonstrate a crossover from 3.5 to $2.5\kappa_0 T$. For the Pfaffian state, $G_Q = 3.5\kappa_0 T$ independent of temperature.

The findings of this work are a notable manifestation of an interplay of equilibration (or absence thereof) and topology in FQH transport. While the charge transport is in the equilibrated regime, the heat transport crosses over from the nonequilibrated to equilibrated regime, with both asymptotic limits characterized by topologically quantized heat conductances determined by edge quantum numbers. We expect that this physics should be relevant also to other FQH states and materials. In particular, interpretation of the experimentally measured thermal conductance $\frac{5}{2}\kappa_0 T$ at the non-Abelian $\nu = 5/2$ state requires assumptions about the presence, absence, or partial character of thermal equilibration[42–46]. Measurement of the full crossover from the nonequilibrated to equilibrated regime would permit to unambiguously resolve this problem.

## Methods

### Device fabrication and measurement scheme

In our experiment, an encapsulated device (heterostructure of hBN/single layer graphene(SLG)/hBN/graphite) was made using the standard dry transfer pickup technique[47]. Fabrication of this heterostructure involved mechanical exfoliation of hBN and graphite crystals on oxidized silicon wafer using the widely used scotch tape technique. First, a hBN of thickness of ~25 nm was picked up at 90 °C using a Poly-Bisphenol-A-Carbonate (PC) coated Polydimethylsiloxane (PDMS) stamp placed on a glass slide, attached to tip of a home built micromanipulator. This hBN flake was aligned on top of previously exfoliated SLG. SLG was picked up at 90 °C. The next step involved the pickup of bottom hBN (~25 nm). This bottom hBN was picked up using the previously picked-up hBN/SLG following the previous process. This hBN/SLG/hBN heterostructure was used to pick up the graphite flake following the previous step. Finally, this resulting hetrostructure (hBN/SLG/hBN/graphite) was dropped down on top of an oxidized silicon wafer of thickness 285 nm at temperature 180 °C. To remove the residues of PC, this final stack was cleaned in chloroform ($CHCl_3$) overnight followed by cleaning in acetone and iso-propyl alcohol (IPA). After this, Poly-methyl-methacrylate (PMMA) photoresist was coated on this heterostructure to define the contact regions in the Hall probe geometry using electron beam lithography (EBL). Apart from the conventional Hall probe geometry, we defined a region of ~5.5 µm² area in the middle of SLG flake, which acts as floating metallic reservoir upon edge contact metallization. After EBL, reactive ion etching (mixture of $CHF_3$ and $O_2$ gas with flow rate of 40 sccm and 4 sccm, respectively at 25 °C with RF power of 60 W) was used to define the edge contact. The etching time was optimized such that the bottom hBN did not etch completely to isolate the contacts from bottom graphite flake, which was used as the back gate. Finally, thermal deposition of Cr/Pd/Au (3/12/60 nm) was done in an evaporator chamber having base pressure of ~$1-2 \times 10^{-7}$ mbar. After deposition, a lift-off procedure was performed in hot acetone and IPA. This results in a Hall bar device along with the floating metallic reservoir connected to the both sides of SLG by the edge contacts. The schematics of the device and measurement setup are shown in Fig. 1d. The distance from the floating contact to the ground contacts was ~5 µm (see Supplementary Fig. 1 for optical images). All measurements were done in a cryo-free dilution refrigerator having a base temperature of ~20 mK. The electrical conductance was measured using the standard lock-in technique, whereas the thermal conductance was measured with noise thermometry based on an LCR resonant circuit at resonance frequency ~740 kHz. The signal was amplified by a home-made preamplifier at 4 K followed by a room temperature amplifier, and finally measured by a spectrum analyzer. Details of the measurement technique are discussed in the Supplementary Fig. 2.

### Description of the crossover from the nonequilibrated to equilibrated regime

When edge modes are not thermally equilibrated, i.e. for edge lengths $L$ satisfying $L \ll \ell_{eq}^H$, the thermal conductance becomes quantized as

$$G_Q = (N_d + N_u)\kappa_0 T, \qquad (3)$$

which means that every edge mode gives a contribution $1\kappa_0 T$ to $G_Q$. For filling factors $\nu = 1/3$, $\nu = 2/5$, $\nu = 2/3$, and $\nu = 3/5$, the corresponding values of the thermal conductance are $G_Q/\kappa_0 T = 1$, 2, 2, and 3, respectively. In fact, the validity of Eq. (3) requires that $L$ also satisfies $L \ll L_T$, where $L_T \sim T^{-1}$ is the thermal length. In the intermediate regime $L_T \ll L \ll \ell_{eq}^H$, a correction to this value emerges due to back-scattering of heat at interfaces with contacts[8,20,33]. For the sake of simplicity, we discard this correction in our analysis in the present work.

In the regime of full thermal equilibration, $L \gg \ell_{eq}^H$, the thermal conductance becomes topologically quantized as

$$G_Q = |N_d - N_u|\kappa_0 T. \tag{4}$$

For $v = 1/3$ and $2/5$ we have $N_u = 0$, so that Eqs. (3) and (4) coincide. For such FQH edges, with only downstream modes, the thermal conductance is thus predicted to be $G_Q = N_d\kappa_0 T$, independent of temperature. This is exactly what is observed in our experiment. On the other hand, for FQH edges with CP modes, i.e., with $N_u > 0$, the equilibrated value (4) is smaller than the nonequilibrated value (3), so that there is a nontrivial crossover of $G_Q$ between the two limits. This is the case for $v = 2/3$ and $v = 3/5$.

For $v = 3/5$, we have $N_d = 1$ and $N_u = 2$, so that $G_Q/\kappa_0 T = 1$. It is worth noting that in this case, $N_d - N_u = -1$, implying that the heat flows upstream on the equilibrated edge, i.e., against the charge flow direction. However, the present experimental setup only measures the absolute value of $G_Q$ and does not reveal the heat flow direction on individual edge segments. The crossover function between the nonequilibrated and equilibrated regime is found to be[8,9,18]

$$\frac{G_Q}{\kappa_0 T} = \frac{2 + e^{-L/\ell_{eq}^H}}{2 - e^{-L/\ell_{eq}^H}} = \frac{2 + e^{-kT^p}}{2 - e^{-kT^p}}, \tag{5}$$

where $L/\ell_{eq}^H = kT^p$. Fitting our experimental data to Eq. (5) with fit parameters $k$ and $p$, we obtain $p \approx 9.34$ (in Fig. 4a).

For the $v = 2/3$ state, we have $N_d = N_u = 1$, so that the equilibrated limiting value of $G_Q$, Eq. (4), is zero. In this case, the crossover takes place between ballistic heat transport in the nonequilibrated regime and heat diffusion in the equilibrated regime[8,9,18]:

$$\frac{G_Q}{\kappa_0 T} = \frac{2\ell_{eq}^H}{L + \ell_{eq}^H} = \frac{2}{1 + kT^p}. \tag{6}$$

Fitting the experimental data to this form, we get the exponent $p \approx 6.34$ (in Fig. 4a).

## Data availability

Additional information related to this work is available from the corresponding author upon reasonable request. Source data are provided with this paper.

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

## Acknowledgements

A.D. thanks the Department of Science and Technology (DST) and Science and Engineering Research Board (SERB), India for financial support (DSTO-2051) and acknowledges the Swarnajayanti Fellowship of the DST/SJF/PSA-03/2018-19. S.K.S. and R.K. acknowledge Prime Minister's Research Fellowship (PMRF), Ministry of Education (MOE) and Inspire fellowship, DST for financial support, respectively. A.D.M. and Y.G. acknowledge support by the DFG Grant MI 658/10-2 and by the German-Israeli Foundation Grant I-1505-303.10/2019. Y.G. acknowledges support by the Helmholtz International Fellow Award, by the DFG Grant RO 2247/11-1, by CRC 183 (project C01), and by the Minerva Foundation. C.S. acknowledges funding from the Excellence Initiative Nano at the Chalmers University of Technology and the 2D TECH VINNOVA competence Center (Ref. 2019-00068). This project has received funding from the European Union's Horizon 2020 research and innovation programme under grant agreement No 101031655 (TEAPOT). K.W. and T.T. acknowledge support from the Elemental Strategy Initiative conducted by the MEXT, Japan and the CREST (JPMJCR15F3), JST.

## Author contributions

S.K.S. and R.K. contributed to device fabrication, data acquisition, and analysis. A.D. contributed in conceiving the idea and designing the experiment, data interpretation, and analysis. K.W. and T.T. synthesized the hBN single crystals. C.S., A.D.M., and Y.G. contributed in development of theory, data interpretation, and all the authors contributed in writing the manuscript.

## Competing interests

The authors declare no competing interests.
