## [Peer Review File · Nature Communications]

REVIEWER COMMENTS

Reviewer #1 (Remarks to the Author):

The manuscript from Saurabh Kumar Srivastav et al. reports on the measurement of thermal Hall conductance of four different fractional quantum Hall (FQH) states in monolayer graphene at different temperatures. The main results are summarized in Figure 4 in the manuscript:

1. At $\nu=1/3$ and $\nu=2/5$, the thermal Hall conductance G_{Q} was measured to be $\approx\kappa_0 T$ and $\approx 2\kappa_0 T$, respectively. These results agree with the quantized values from theoretical prediction. Furthermore, the measured G_{Q} were temperature independent.

2. At $\nu=2/3$ and $\nu=3/5$, the measured values of G_{Q} showed a crossover respectively from $2\kappa_0 T$ and $3\kappa_0 T$ to $0.5\kappa_0 T$ and $\kappa_0 T$ as the temperature of the system was increased.

Combining the above observation, the work has demonstrated the fundamental difference between heat transport in quantum Hall edges with and without counter-propagating edge modes. In addition, the existence of crossover in hole-like FQH states is consistent with the concept of thermal equilibration length, which is itself a temperature dependent quantity.

In my opinion, the experiment was performed systematically. The experimental procedures and theoretical analysis have been clearly explained in the main text and supplementary information. The results are also consistent with most of the existing theoretical predictions. As the authors have correctly mentioned, a similar crossover in thermal Hall conductance was observed in GaAs heterostructures. However, the possible degree of freedoms and microscopic details of graphene are quite different from GaAs. For the four FQH states studied in this work, the results demonstrate the universality of their edge structures in both GaAs and graphene samples. I believe this is an important step in studying FQH states in graphene samples. Therefore, I would recommend a publication of the manuscript in Nature Communications.

To make the manuscript more comprehensive, I think some of the theoretical discussions can be further elaborated:

1. Edge reconstruction commonly occurs in hole-like FQH states in GaAs heterostructures. In principle, this may also occur in graphene samples. Then, the interpretation of $G_{Q}/\kappa_0 T$ becomes more subtle in the non-equilibrated regime. Meanwhile, $G_{Q}/\kappa_0 T = |N_d - N_u|/\kappa_0 T$ is still satisfied if the reconstructed edge is fully equilibrated. From the crossover, it seems that edge reconstruction does not occur in the present graphene sample. Can the authors briefly comment on this issue? Is edge reconstruction always avoided in graphene samples due to the sharp confining potential?

2. Although it is now commonly believed that there should have upstream mode(s) in the $\nu=2/3$ edge, it was originally believed that the $2/3$ edge has two copropagating $1/3$ modes. This would of course lead to $G_{Q}/\kappa_0 T = 2$ independent of the temperature. The observation of crossover in $G_{Q}/\kappa_0 T$ basically eliminates this $1/3-1/3$ picture in the graphene sample, and supports the $1-(-1/3)$ picture as in GaAs heterostructure without edge reconstruction. I think the authors may discuss this simple example to further motivate how can the observation of crossover in $G_{Q}/\kappa_0 T$ determine the topological number of the FQH edges, as claimed in the title of the manuscript.

3. At the end of the section "Discussion", the authors have claimed that the large values of ξ may originate from multiparticle processes or/and nonlinear edge spectrum. Is it because these mechanisms involve more irrelevant process (in the language of renormalization group). Therefore, the temperature dependence of thermal equilibration length will have a larger exponent after some simple RG analysis? Can the authors explain briefly on this or provide some references?

4. As the authors pointed out in the manuscript, the present work may provide the first step in understanding the exotic non-Abelian quantum Hall states, such as the FQH state at $\nu=5/2$. It may be useful and more motivating if the authors can discuss this issue a little bit further. For example, anti-Pfaffian state will demonstrate a crossover from $4.5\kappa_0 T$ to $1.5\kappa_0 T$, whereas the PH-Pfaffian state will demonstrate a crossover from $3.5\kappa_0 T$ to $2.5\kappa_0 T$. For Pfaffian state (without edge reconstruction), $G_{Q}/\kappa_0 T = 3.5$ independent of temperature.

Finally, there are some other minor suggestions:

1. Following point 2 above, I would suggest the title of the manuscript should be more specific. It should state clearly thermal Hall conductance measurement was employed.

2. In the caption of Figure 1, "Robust fractional plateaus at $e^2/3h$, $2e^2/5h$, $3e^2/5h$, and $2e^2/3h$ are clearly visible". I think it should be revised to $3h/e^2$, $5h/2e^2$, $5h/3e^2$, and $3h/2e^2$. It is because Fig. 1(e) shows the Hall resistance but not the Hall conductance. The revision will make the caption more consistent with the figure.

3. In the Introduction, the authors mentioned the base temperature was $\sim 20\text{mK}$. It will be helpful if the authors can clarify what does this base temperature mean. Does it mean the temperature of the sample? Later, the bath temperature T_{bath} enters in the discussion. I think this is not the same as the base temperature, am I right?

4. There is a typo above Eq.(S7), G_{j} should be G_{i} .

Reviewer #2 (Remarks to the Author):

The same authors have a related work recently (reference 22 of the current manuscript). The manuscript used similar techniques and extended the measurement from the previous $2/3$ to other states, and discussed the different temperature behaviors from different states. I am not convinced that the additional contribution from this manuscript deserves another publication in NATURE COMMUNICATIONS.

As a reader, I am interested in what may happen if the data in Fig4a could be extended to higher temperature and I would also be interested in more discussion with their own data from reference 22.

Reviewer #3 (Remarks to the Author):

The present paper identifies quantum numbers associated with quantum Hall edges at different filling fractions by probing a transition in thermal conductance from the equilibrated to the non-equilibrated regimes. These quantum numbers are useful for understanding the structure of these edge states, in particular, how many neutral modes and upstream modes they possess. From these

one can learn about the role of interactions and edge reconstruction. It has also been recently appreciated that equilibration plays an important role in determining the measured quantum numbers. Understanding the role of equilibration is important fundamentally, to understand how charge and energy are redistributed in interacting edges. It also has application to understand measurements of half-quantized edge states which may possess non-Abelian quasiparticles, since a half-quantized measurement may in actuality be a partially-equilibrated edge. Understanding equilibration is therefore highly important to the field.

The present experiment claims to achieve an important result, which is the in situ observation of a transition in the equilibration regimes of edge states possessing upstream neutral modes. This transition is observed for two hole-like states, where the transition is expected, but not for particle-like states, where it is not. To achieve this observation, the authors have utilized the method of thermal conductance measurement first demonstrated by Jezouin, et al., and subsequently extended to fractional, non-Abelian, and more recently graphene quantum Hall states. Broadly, the authors appear to have applied this technique in a reasonably careful manner. On the basis of the above, the paper potentially represents an important contribution to the field and, in principle, could be published in Nature Communications. However, there are some issues that need to be addressed before this reviewer can recommend publication.

A major issue concerns the way in which the results are discussed and written. The section labelled 'Results' contains statements of interpretation that are treated as known facts. For example, the sentence "On the other hand...", which includes the first description of the hole-like states data, immediately treats the data as the crossover from non-equilibration to equilibration. The proper approach is to present the key features of the data without interpretation, and following this the authors can invoke a model of choice to demonstrate consistency with the data. The model they have chosen appears only in the Methods section; the model should be introduced and discussed directly in the Results section, separately from first discussing the key features of the data (a more detailed discussion of the model can remain in the Methods section). The final two sentences of this section, which make claims about a first observation or which invoke states (even denominator, $5/2$) which are not observed here, should be removed, as they are not appropriate in a discussion of scientific results.

An improved approach to discussing the results would better frame the strengths of the data, namely its consistency with two expected regimes, while clarifying those features that are unexplained and which may be at odds with the offered interpretation:

- First, the large value of the temperature exponent p which was not expected. Possible explanations are mentioned but no references or more significant discussion are offered.
- Second, the absence of a second transition as predicted in certain theories. Is this allowed by those theories or does it imply a different scenario?

- Third, the temperature window in which the transition occurs is not discussed, and it is unclear whether this temperature scale is predicted by any existing theory or otherwise explainable. Can the authors provide any insight into this issue?

Further questions are below:

- The method of Jezouin et al. relies on a floating metal contact to serve as a bath that generates a consistent electron temperature and which does not generate additional shot noise. It is unclear whether this assumption is upheld by the device in this work:

o The width of the floating metal contact appears to be 1 micrometer. What is the traversal time and expected thermalization time for a thin metal film of the deposited type at this temperature?

o The authors do not appear to have checked whether the reflection coefficient from the floating metal contact is sufficiently small to avoid shot noise contributions to the measured signal (see i.e. Banerjee, et al., 2017). Why is this the case? Is such data available and consistent with the requirements?

o For both of the above points, can it be ruled out that a change in these properties of the floating contact as a function of bath temperature would lead to the observed behavior?

- The authors have not presented the temperature dependence of the resistance at the measured filling fractions. (Only two temperature points are shown in a gate voltage sweep in Fig.S5). Is this data available? This would help rule out a change in electrical resistance as the origin of a measured thermal conductance change. For this purpose it would also be helpful to know the specific gate voltage associated with the filling fractions labelled in the figures, which is missing.

- No temperature dependence of integer states has been presented. Is such data available? This would be an important check as all the presented data is in the fractional regime.

- Why is the bath temperature sweep limited from 20 to 60mK? This seems a narrow window, particularly as the $2/3$ state has not attained its asymptotic value, which would be of high interest. If there is not a hard experimental limit at 60mK, can the window be extended?

Reviewer # 1 (Remarks to the author):

The manuscript from Saurabh Kumar Srivastav et al. reports on the measurement of thermal Hall conductance of four different fractional quantum Hall (FQH) states in monolayer graphene at different temperatures. The main results are summarized in Figure 4 in the manuscript:

1. At $\nu=1/3$ and $\nu=2/5$, the thermal Hall conductance G_Q was measured to be $\approx \kappa_0 T$ and $\approx 2\kappa_0 T$, respectively. These results agree with the quantized values from theoretical prediction. Furthermore, the measured G_Q were temperature independent.
2. At $\nu=2/3$ and $\nu=3/5$, the measured values of G_Q showed a crossover respectively from $2\kappa_0 T$ and $3\kappa_0 T$ to $0.5\kappa_0 T$ and $\kappa_0 T$ as the temperature of the system was increased.

Combining the above observation, the work has demonstrated the fundamental difference between heat transport in quantum Hall edges with and without counter-propagating edge modes. In addition, the existence of crossover in hole-like FQH states is consistent with the concept of thermal equilibration length, which is itself a temperature dependent quantity.

In my opinion, the experiment was performed systematically. The experimental procedures and theoretical analysis have been clearly explained in the main text and supplementary information. The results are also consistent with most of the existing theoretical predictions. As the authors have correctly mentioned, a similar crossover in thermal Hall conductance was observed in GaAs heterostructures. However, the possible degree of freedoms and microscopic details of graphene are quite different from GaAs. For the four FQH states studied in this work, the results demonstrate the universality of their edge structures in both GaAs and graphene samples. I believe this is an important step in studying FQH states in graphene samples. Therefore, I would recommend a publication of the manuscript in Nature Communications.

We thank the Reviewer for an excellent summary of our work. We are very pleased to learn that the Reviewer finds our results an important step in the study and characterization of FQH states in graphene. Furthermore, we are very pleased to read that the Reviewer points to the agreement of theoretical predictions with the thorough and systematic experimental study. The reviewer correctly mentions that this study on graphene, with additional degrees of freedom and further microscopic details as compared with GaAs, demonstrates universal facets of the edge structure. To further emphasize the novelty of our result, it should be noted that for GaAs, the two asymptotic limits of the thermal conductance were only observed in separate distinct devices; by contrast, the crossover from one limiting case to another, is demonstrated here for the first time. In summary, we are delighted to read the positive recommendation of the Reviewer for publication in Nature Communications.

To make the manuscript more comprehensive, I think some of the theoretical discussions can be further elaborated:

We are further grateful for Reviewer's encouragement to make the manuscript more comprehensive; this step has indeed been taken in the revised manuscript.

1. Edge reconstruction commonly occurs in hole-like FQH states in GaAs heterostructures. In principle, this may also occur in graphene samples. Then, the interpretation of G_Q becomes more subtle in the non-equilibrated regime. Meanwhile, $G_Q = |N_d - N_u| \kappa_0 T$ is still satisfied if the reconstructed edge is fully equilibrated. From the crossover, it seems that edge reconstruction does not occur in the present graphene sample. Can the authors briefly comment on this issue? Is edge reconstruction always avoided in graphene samples due to the sharp confining potential?

We thank Reviewer 1 for raising this important point. Indeed, it has been proposed that edge reconstruction might occur in various QH states (with and without counter propagating modes, fractional and integer). In fact, measurements of the thermal conductance, as performed in our work, is a perfect tool to probe whether it takes place. For example, for the $\nu=1/3$ state, edge reconstruction would increase the number of modes from 1 to 3. This implies that at low temperatures (non-equilibrated regime) the value of the thermal conductance would be $G_Q = 3\kappa_0 T$, with a crossover to $G_Q = 1\kappa_0 T$ (equilibrated regime) at higher temperature. This is not observed in our experiment: the conductance is $G_Q = 1\kappa_0 T$ down to the lowest temperatures. Similarly, for the $\nu=2/3$ state, edge reconstruction would increase the number of modes from 2 to 4. This implies that at low temperatures (non-equilibrated regime) the value of the thermal conductance would be $G_Q = 4\kappa_0 T$. Again, no indication of this is observed, implying no edge reconstruction in our device. This important point has been highlighted in the revised manuscript.

The edge reconstruction is a result of competition between the confinement potential that holds the electrons in the interior of the sample, and the Coulomb repulsion that tends to spread out the electron density. It is useful to note that edge reconstruction may take place also for particle-like Laughlin states (cf. e.g., the appearance of upstream modes in GaAs samples (H. Inoue et al., Nat. Comm. 5, 4067 (2014)) and a general theoretical study (U. Khanna et al., Phys. Rev. B 103, L121302 (2021)). In fact, it has been theoretically predicted (Zi-Xiang Hu et al, PRL 107, 236806 (2011), for the Laughlin like states) that edge reconstruction in graphene samples can be avoided, provided the separation between the graphene and the screening layer (metallic gate - d) and the magnetic length scale ($l_B = \sqrt{\hbar/eB}$) are comparable ($0.5l_B \leq d \leq 1.5l_B$). Since, for our device, the hBN thickness (~ 25 nm) separating the graphene and the bottom graphite gate is very close to the limit of mentioned criteria, one can in principle, expect that edge reconstruction may not take place for our device, which is indeed experimentally verified by our thermal conductance results. We believe that it is also in accordance with the STM experiment performed for monolayer graphene supported on graphite [Nature Communications 4,1744 (2013)], where the authors did not observe any sign of edge reconstruction for integer quantum Hall states. However, it is worth to mention here that the edge reconstruction has also been reported in a local

scanning gate microscope experiment [A. Marguerite et al, Nature 575, 628–633(2019)] where the hBN encapsulated monolayer graphene device was supported on a SiO_2/Si substrate, and hence the graphene channel was separated by ~ 300 nm SiO_2 layer from metallic gate (highly doped Si). This distance is much larger than the theoretical limit quoted above from Zi-Xiang Hu et al, PRL 107, 236806 (2011). Together, these reports indeed suggest that edge reconstruction can be avoided in graphene if the confining potential is sharp. However, more experimental and theoretical studies are required to fully determine this issue.

Following the recommendation of the Reviewer, we have included a discussion of the edge reconstruction in the Section “Introduction” and emphasized that our results imply the absence of edge reconstruction in the Section “Results” of the revised manuscript.

2. Although it is now commonly believed that there should have upstream mode(s) in the $\nu=2/3$ edge, it was originally believed that the $2/3$ edge has two co-propagating $1/3$ modes. This would of course lead to $G_Q=2\kappa_0T$ independent of the temperature. The observation of crossover in G_Q basically eliminates this $1/3$ - $1/3$ picture in the graphene sample, and supports the 1 - $(-1/3)$ picture as in GaAs heterostructure without edge reconstruction. I think the authors may discuss this simple example to further motivate how can the observation of crossover in G_Q determine the topological number of the FQH edges, as claimed in the title of the manuscript.

We thank the Reviewer for guiding us to make our manuscript more comprehensive. We completely agree with the Reviewer that our experimental observation of the crossover in G_Q with decreasing temperature confirms that the edge structure of $2/3$ for graphene supports the theoretical model of 1 - $(-1/3)$ picture, similar to GaAs, and rules out the other theoretical model of $1/3$ $+1/3$ picture. As suggested by the Reviewer, in the revised version, we have therefore included a new section “Edge structure and thermal conductance” describing different proposed theoretical models for the hole-conjugate fractional quantum Hall states and the corresponding expected thermal conductance. We fully agree with the Referee that this helps to demonstrate in the most clear way how the observation of crossover in G_Q determines exact topological numbers of the FQH edges.

3. At the end of the section "Discussion", the authors have claimed that the large values of ρ_{xx} may originate from multiparticle processes or/and nonlinear edge spectrum. Is it because these mechanisms involve more irrelevant process (in the language of renormalization group). Therefore, the temperature dependence of thermal equilibration length will have a larger exponent after some simple RG analysis. Can the authors explain briefly on this or provide some references?

The Luttinger-liquid model of FQH edges is an integrable theory and thus does not involve relaxation. The relaxation comes from additional terms that break integrability. This may be, in particular, (i) intermode tunnelling operators or (ii) non-linear corrections to spectra of fermionic excitations and of plasmons. The Reviewer is perfectly correct that higher values of

the exponent ν of the equilibration length correspond to processes that are more irrelevant in the renormalization-group sense. This is the case for more complex (multiparticle) tunnelling as well as for processes related to nonlinearities. To find which of such competing processes would be the most important ones on the FQH edge is however a challenging task, as this is expected to depend on microscopic parameters and of the temperature range. This analysis is beyond the scope of this work. Following the suggestion by the Referee, we have added in section “Discussion” of the revised manuscript (i) a comment clarifying that the equilibration is due to RG-irrelevant processes and (ii) several references on mechanisms of the equilibration in strongly correlated 1D systems that may lead to large ν :

C.L. Kane and Matthew P.A. Fisher, Phys. Rev. B. **46** 15233 (1992)

C.L. Kane and Matthew P.A. Fisher, Phys. Rev. B. **51** 13449 (1995)

X.-G. Wen: Adv. Phys. **44**, 405 (1995)

J. T. Chalker, Y. Gefen, and M. Y. Veillette, Phys. Rev. B 76, 085320 (2007)

Imambekov, A., Schmidt, T. L. & Glazman, L. I., Rev. Mod. Phys. 84, 1253–1306 (2012).

Protopopov, I. V., Gutman, D. B. & Mirlin, A. D. , Phys. Rev. B 90, 125113 (2014).

Apostolov, S., Liu, D. E., Maizelis, Z. & Levchenko, A., Phys. Rev. B 88, 045435 (2013).

Lin, J., Matveev, K. A. & Pustilnik, M., Phys. Rev. Lett. 110, 016401 (2013).

Arzamasovs, M., Bovo, F. & Gangardt, Phys. Rev. Lett. 112, 170602 (2014).

Protopopov, I. V., Gutman, D. B. & Mirlin, A. D. , Phys. Rev. B 91, 195110 (2015).

4. As the authors pointed out in the manuscript, the present work may provide the first step in understanding the exotic non-Abelian quantum Hall states, such as the FQH state at $\nu=5/2$. It may be useful and more motivating if the authors can discuss this issue a little bit further. For example, anti-Pfaffian state will demonstrate a crossover from $4.5\kappa_0 T$ to $1.5\kappa_0 T$, whereas the PH-Pfaffian state will demonstrate a crossover from $3.5\kappa_0 T$ to $2.5\kappa_0 T$. For Pfaffian state (without edge reconstruction), $G_Q=3.5\kappa_0 T$ independent of temperature.

Once again we thank the Reviewer for emphasizing the importance of our work in the context of understanding the exact topological phase of non-Abelian quantum Hall states by measuring the crossover of the thermal conductance. In the revised manuscript, we have added, following the recommendation of the Reviewer, a paragraph in the discussion section with the expected thermal conductance values and their crossover behaviour for different possible ground states of the $\nu=5/2$ state.

Finally, there are some other minor suggestions:

1. Following point 2 above, I would suggest the title of the manuscript should be more specific. It should state clearly thermal Hall conductance measurement was employed.

Base on the Reviewer’s suggestion, we have changed the title to “Determination of topological edge quantum numbers of fractional quantum Hall phases by thermal conductance measurements.”

2. In the caption of Figure 1, "Robust fractional plateaus at $e^2/3h$, $2e^2/5h$, $3e^2/5h$, and $2e^2/3h$ are clearly visible". I think it should be revised to $3h/e^2$, $5h/2e^2$, $5h/3e^2$, and $3h/2e^2$. It is

because Fig. 1(e) shows the Hall resistance but not the Hall conductance. The revision will make the caption more consistent with the figure.

As suggested by the Reviewer, we have incorporated these changes in the revised manuscript.

3. In the Introduction, the authors mentioned the base temperature was $\sim 20\text{mK}$. It will be helpful if the authors can clarify what does this base temperature mean. Does it mean the temperature of the sample? Later, the bath temperature T_0 enters in the discussion. I think this is not the same as the base temperature, am I right?

We thank the Reviewer for this important comment. The base temperature is the lowest bath temperature ($T_{\text{bath}} \sim 20\text{mK}$) achieved in our setup. However, T_0 refers to the electron temperature of the cold grounded (CG) contacts of the sample (see Fig. 1) which, in principle, can be different from the bath temperature (T_{bath}). In our experiment, we have carefully estimated the electron temperature (T_0) for each value of the bath temperature (T_{bath}). The details of this estimation procedure is discussed in the SI in section S2. The corresponding plots are also shown in Fig. S3. It should be noted that above $T_{\text{bath}} \geq 20\text{ mK}$, $T_0 \approx T_{\text{bath}}$. We have clarified the different temperatures in the revised manuscript.

4. There is a typo above Eq.(S7), $G_{j \leftarrow i}$ should be $G_{j \leftarrow i}$.

We have corrected this typo in the revised supplementary file.

Reviewer # 2 (Remarks to the author):

The same authors have a related work recently (reference 22 of the current manuscript). The manuscript used similar techniques and extended the measurement from the previous 2/3 to other states, and discussed the different temperature behaviours from different states. I am not convinced that the additional contribution from this manuscript deserves another publication in NATURE COMMUNICATIONS.

We thank the Reviewer for going through our manuscript. But here we respectfully disagree with the Reviewer's comment that this work is just the extension of the previous paper on 2/3 state [Kumar et al, Nature Comm., 13, 213 (2022), which was Ref. 22 of the original version] to other states. We would therefore like to point out the following *major* differences between the two experiments:

1. The current manuscript studies the thermal conductance of particle-like and hole-like fractional quantum Hall states while our earlier work investigated the noise, i.e., a very different physical characteristics of the system. Furthermore, observation of the crossover of the thermal conductance between two limits (non-equilibrated and

equilibrated) offers an unambiguous determination of the exact topological edge quantum numbers of the fractional quantum Hall states. This is in contrast to the earlier work (Ref. 22 of the original version) which only demonstrated the presence of the upstream modes. That work did not reveal anything about the number of the topological edge modes—a key information needed to determine the exact topological order of the ground state of fractional quantum Hall states. In the revised manuscript, we have added a new section “Edge structure and thermal conductance” describing different proposed theoretical models for the hole-conjugate fractional quantum Hall states and the corresponding expected thermal conductance and crossovers. This further underscores how the experimental observation of crossover in G_Q , presented in the current manuscript for the first time, divulges the precise topological numbers of the fractional quantum Hall phases.

2. Secondly, the current manuscript not only comprises an important step in determining the detailed edge structure; it also rules out any possible edge reconstruction in the presented graphene device. For example, for the $\nu=2/3$ state, edge reconstruction would increase the number of modes from 2 to 4. This implies that at low temperatures (non-equilibrated regime) the value of the thermal conductance would be $G_Q = 4\kappa_0 T$ instead of the observed value of $2\kappa_0 T$. No indication of this is observed, implying no edge reconstruction in our device. In the revised manuscript, we have highlighted how the thermal conductance measurement can unambiguously tell whether there is any edge reconstruction or not. By contrast, the previous experiment (Ref. 22) only discloses the presence of upstream modes but cannot tell much about the edge reconstruction issue. Furthermore, our experiment paves the way towards the understanding of the exact topological order of more complex states like $5/2$ state in graphene, whose ground state is still unknown. For example, the anti-Pfaffian state should exhibit a crossover from $4.5\kappa_0 T$ to $1.5\kappa_0 T$ whereas the PH-Pfaffian state should exhibit a crossover from $3.5\kappa_0 T$ to $2.5\kappa_0 T$, and for the Pfaffian state $G_Q = 3.5\kappa_0 T$ independently of the temperature. This has been highlighted in the revised manuscript in the discussion section.
3. Additionally, it is noteworthy pointing out major differences in the device configuration and the measurement scheme between the current experiment and Ref. 22. In the current work, we measure the thermal conductance, for which one needs a set of hot and cold reservoirs, and by measuring the steady state temperature of the hot reservoir, which depends on the thermal conductance, we extract the value of G_Q . To create the hot reservoir, in the current manuscript, we have used a metallic floating contact, which is the heart of the measurement scheme, in the middle of the device [see Fig.1(d)], and was not present in Ref. 22. To determine the temperature, we measure the excess thermal noise coming out from the hot reservoir in the current experiment, which is directly related to the electron temperature of the metallic floating contact and does not depend on the heat leakage from the quantum Hall edges to the environment. Thus, the crossover of G_Q is purely coming from the equilibration on the edge. On the contrary, in our previous work (Ref. 22), the dissipated

power from injecting a current creates a “hot spot” at the back of a source contact, and due to the presence of upstream modes, the heat is transported upstream to the so-called “noise spot”. At this spot, the heat partitions the charge current and thereby generates partitioning noise. In this case, the resulted noise was sensitive to the heat leakage from the quantum Hall edges to the environment. Thus, as was explicitly pointed out in Ref. 22, it was not clear whether the observed decay of the noise there was because of equilibration between the edge modes or because of the heat leakage to the environment. Motivated by the comment by the Referee, we have now emphasized all these important points (1, 2, and 3) in the revised version of the manuscript.

As a reader, I am interested in what may happen if the data in Fig4a could be extended to higher temperature.

As pointed out above, the current experiment involves a metallic floating contact, which is heated to a temperature T_M . As a result, the heat current flows along the ballistic edge channel between this hot floating metallic contact and cold reservoirs. As mentioned above, the heat leakage from the quantum Hall edges to the environment does not affect the determination of the thermal conductance. At the same time, the heat loss of the floating contact via electron-phonon cooling at an elevated temperature does affect the energy balance on the island: the heat that leaves the island via the edge is then not equal to the dissipated Joule heat. This affects the determination of the thermal conductance value in an essential manner. We have found that for our graphene devices the electron-phonon contribution starts to influence substantially our measurement at floating-contact temperatures (T_M) around 95-100 mK. This can be seen from figure S 12 in the SI. Due to this effect, we have restricted our measurement to the bath temperature (T_{bath}) of 60 mK, where the T_M is increased to around 70-80 mK in order to fit a sufficient range of data for the G_Q extraction (Fig. 3 in the main manuscript). One could increase T_{bath} further by 5 to 10 mK, but it will not affect the central result of the current work as for 3/5 we have already seen the thermal conductance plateau for the equilibrated regime, and for 2/3 an asymptotic decay to $\sim 0.5 \kappa_0 T$ for the diffusive nature. Observing the asymptotic value of 2/3 reaching $\approx 0 \kappa_0 T$ will be very interesting at the elevated temperature, but that will be really difficult to achieve for a length of $\sim 5 \mu\text{m}$ channel, as the electron-phonon coupling affects the measurement very significantly at such temperatures. It should be noted that with suitable longer channel length one could achieve this goal. It is also worth noting that even for $\sim 150 \mu\text{m}$ channel length in GaAs, Banerjee, et al., 2017, reported $G_Q \sim 0.33 \kappa_0 T$ for a 2/3 edge at 10mK.

For the convenience of the Reviewer, we are including SI figure S 12 (below) with this response to emphasize why there is a limitation in thermal conductance measurement at temperature beyond $T_{\text{bath}} > 60\text{mK}$.

In response to this question by the Reviewer #2 and a similar question by the Reviewer #3, we have added in the revised version an explanation on the upper border of our temperature range.

Figure: Top panel; left: Thermal current J_Q is plotted as function of $T_M^2 - T_0^2$ for $\nu = 1$ at $T_{\text{bath}} = 20$ mK (corresponding electron temperature $T_0 \sim 23$ mK). The deviation from the linearity is attributed to the electron-phonon cooling. Right: The electronic contribution of thermal current is subtracted from the total thermal current J_Q which is then plotted as function of $T_M^5 - T_0^5$. Beyond the position of vertical dashed line, which correspond to $T_M \sim 100$ mK of the floating contact, the electron-phonon cooling becomes significant.

Bottom panel: Same plots at $T_{\text{bath}} = 50$ mK (corresponding electron temperature $T_0 \sim 52$ mK). Here, also one can see that beyond $T_M \sim 96$ mK of the floating contact, the electron-phonon cooling becomes significant.

I would also be interested in more discussion with their own data from reference 22.

We understand that the Reviewer asks about a comparison of results of this work with those of Ref. 22 (numbering follows the original version of this manuscript). We recall that the physical observables measured in these two works are different (thermal conductance here vs noise in Ref. 22). We thus conclude that what can be compared are conclusions concerning the edge structure. It was shown in Ref. 22 that there are upstream modes in 2/3 and 3/5 states, whereas no such modes is found in 1 and 1/3 edges. Furthermore, at low temperatures the noise in 2/3 and 3/5 states was found to be L-independent in Ref. 22, implying the ballistic (non-equilibrated) character of the thermal transport. These finding are in full agreement with the conclusions of the present work (based on thermal-conductance measurements). While such an agreement between two distinct experiments is very important by itself, the present work (thermal-conductance study) allows us to go much further. By observing the equilibrated and non-equilibrated regimes in one sample, we can directly extract the quantum numbers N_u and N_d characterizing the edge from the value of the thermal

conductance. This also allows us to demonstrate that there no edge reconstruction takes place in our graphene device.

The comparison to Ref. 22 that appeared in the previous version (Ref. 30 of the revised version), is now presented in the section “Results” of the revised version.

Reviewer # 3 (Remarks to the author):

The present paper identifies quantum numbers associated with quantum Hall edges at different filling fractions by probing a transition in thermal conductance from the equilibrated to the non-equilibrated regimes. These quantum numbers are useful for understanding the structure of these edge states, in particular, how many neutral modes and upstream modes they possess. From these one can learn about the role of interactions and edge reconstruction. It has also been recently appreciated that equilibration plays an important role in determining the measured quantum numbers. Understanding the role of equilibration is important fundamentally, to understand how charge and energy are redistributed in interacting edges. It also has application to understand measurements of half-quantized edge states which may possess non-Abelian quasiparticles, since a half-quantized measurement may in actuality be a partially-equilibrated edge. Understanding equilibration is therefore highly important to the field.

We thank the Reviewer for this accurate and concise summary of our work. We are very pleased to learn that the Reviewer finds our results significant for the field, not the least as a method to account for thermal conductance results of the exotic $5/2$ state.

The present experiment claims to achieve an important result, which is the in-situ observation of a transition in the equilibration regimes of edge states possessing upstream neutral modes. This transition is observed for two hole-like states, where the transition is expected, but not for particle-like states, where it is not. To achieve this observation, the authors have utilized the method of thermal conductance measurement first demonstrated by Jezouin, et al., and subsequently extended to fractional, non-Abelian, and more recently graphene quantum Hall states. Broadly, the authors appear to have applied this technique in a reasonably careful manner. On the basis of the above, the paper potentially represents an important contribution to the field and, in principle, could be published in Nature Communications. However, there are some issues that need to be addressed before this reviewer can recommend publication.

We thank the Reviewer for this very positive assessment of our work. We are delighted to read the following comments “The present experiment claims to achieve an important result” and “the paper potentially represents an important contribution to the field and, in principle, could be published in Nature Communications.”

A major issue concerns the way in which the results are discussed and written. The section labelled 'Results' contains statements of interpretation that are treated as known facts. For example, the sentence "On the other hand...", which includes the first description of the hole-like states data, immediately treats the data as the crossover from non-equilibration to equilibration. The proper approach is to present the key features of the data without interpretation, and following this the authors can invoke a model of choice to demonstrate consistency with the data. The model they have chosen appears only in the Methods section; the model should be introduced and discussed directly in the Results section, separately from first discussing the key features of the data (a more detailed discussion of the model can remain in the Methods section). The final two sentences of this section, which make claims about a first observation or which invoke states (even denominator, 5/2) which are not observed here, should be removed, as they are not appropriate in a discussion of scientific results.

We thank the Reviewer for this suggestion which will indeed improve the quality of the presentation of our work. Based on the Reviewer's input we have made the following changes in the structure of the revised manuscript.

(1) We have introduced a new section "Edge structure and thermal conductance" describing different proposed theoretical models for the fractional quantum Hall states and the corresponding expected thermal conductance values at different regimes (as also proposed by Reviewer #1). For example, for $2/3$, the theoretical model of $1-(-1/3)$ picture will give a crossover with temperature from a thermal conductance value of $2\kappa_0 T$ to $\approx 0\kappa_0 T$, whereas the theoretical model of $1/3 + 1/3$ will lead to $2\kappa_0 T$ independent of the temperature. Similarly, we give the expected theoretical values of G_Q for other FQH states.

(2) In the revised version, in the section "Results", we first describe the key features of our experiment. We then invoke the theoretical model described in the "Edge structure and thermal conductance" section to explain our observations.

(3) We introduce the model for extracting the temperature exponent in the section "Results".

(4) As suggested by the Reviewer, we have removed the last two sentences from the section "Results" of the original version. The discussion of a prospective extension to the $5/2$ state is now included in the section "Discussion".

An improved approach to discussing the results would better frame the strengths of the data, namely its consistency with two expected regimes, while clarifying those features that are unexplained and which may be at odds with the offered interpretation:

The exposition in the revised manuscript follows the suggestion of the Reviewer. In the Results section, we first present the experimental results. After this, we compare them to the theoretical model (the paragraph starting "To understand these results" and explain how the topological quantum numbers of the edge are extracted. Then we point out unexplained features like high temperature exponent; we offer plausible explanations in the subsequent Discussion section.

- First, the large value of the temperature exponent p which was not expected. Possible explanations are mentioned but no references or more significant discussion are offered.

This recommendation is similar to one of recommendations of Referee #1. For convenience, we repeat the response here.

The Luttinger-liquid model of FQH edges is an integrable theory and thus does not involve relaxation. The relaxation comes from additional terms that break integrability. This may be, in particular, (i) intermode tunnelling operators or (ii) non-linear corrections to spectra of fermionic excitations and of plasmons. The Reviewer is perfectly correct that higher values of the exponent p of the equilibration length correspond to processes that are more irrelevant in the renormalization-group sense. This is the case for more complex (multiparticle) tunnelling as well as for processes related to nonlinearities. To find which of such competing processes would be the most important ones on the FQH edge is however a challenging task, as this is expected to depend on microscopic parameters and of the temperature range. This analysis is beyond the scope of this work. Following the suggestion by the Referee, we have added in section “Discussion” of the revised manuscript (i) a comment clarifying that the equilibration is due to RG-irrelevant processes and (ii) several references on mechanisms of the equilibration in strongly correlated 1D systems that may lead to large p :

C.L. Kane and Matthew P.A. Fisher, Phys. Rev. B. **46** 15233 (1992)

C.L. Kane and Matthew P.A. Fisher, Phys. Rev. B. **51** 13449 (1995)

X.-G. Wen: Adv. Phys. **44**, 405 (1995)

J. T. Chalker, Y. Gefen, and M. Y. Veillette, Phys. Rev. B **76**, 085320 (2007)

Imambekov, A., Schmidt, T. L. & Glazman, L. I. , Rev. Mod. Phys. **84**, 1253–1306 (2012).

Protopopov, I. V., Gutman, D. B. & Mirlin, A. D. , Phys. Rev. B **90**, 125113 (2014).

Apostolov, S., Liu, D. E., Maizelis, Z. & Levchenko, A., Phys. Rev. B **88**, 045435 (2013).

Lin, J., Matveev, K. A. & Pustilnik, M., Phys. Rev. Lett. **110**, 016401 (2013).

Arzamasovs, M., Bovo, F. & Gangardt, Phys. Rev. Lett. **112**, 170602 (2014).

Protopopov, I. V., Gutman, D. B. & Mirlin, A. D. , Phys. Rev. B **91**, 195110 (2015).

- Second, the absence of a second transition as predicted in certain theories. Is this allowed by those theories or does it imply a different scenario?

We suppose that the Referee asks here about possible intermediate plateaus in G_Q , between the fully non-equilibrated and fully equilibrated regimes. There are different possible reasons for such plateaus that have been discussed in theoretical papers; we are not sure what exactly the Reviewer has in mind. One of these reasons is a partial equilibration. For example, if one has an edge with 2 upstream and 2 downstream modes, then one may have a pattern $4\kappa_0 T \rightarrow 2\kappa_0 T \rightarrow 0\kappa_0 T$. This requires, however, at least two upstream and two downstream modes, which is not the case with the fractions that we study. This would happen with the $2/3$ state if it would experience edge reconstruction; our results show that it does not happen, as is explained in the revised version of the paper. (See also our response to Reviewer #1.) Another possible reason for an intermediate plateau is a backscattering of heat at contacts. We may speculate that this plateau is not resolved in our device since the non-equilibrated-to-equilibrated crossover is rather sharp.

- Third, the temperature window in which the transition occurs is not discussed, and it is unclear whether this temperature scale is predicted by any existing theory or otherwise explainable. Can the authors provide any insight into this issue?

The temperature scale for the crossover in G_Q is highly non-universal. The crossover occurs when the system size L becomes comparable to the thermal equilibration length: $L \sim l_{eq}$. The equilibration length depends on the mechanism of the equilibration (see above) and of the corresponding parameters (such as the tunnelling amplitude, interaction strength, non-linearities). So, it is difficult to make a theoretical prediction of the value of l_{eq} (and thus of the crossover temperature).

The width of the crossover window depends on the exponent β (the larger β the sharper is the crossover). This is now discussed in the manuscript (see the response to the corresponding question above)

- The method of Jezouin et al. relies on a floating metal contact to serve as a bath that generates a consistent electron temperature and which does not generate additional shot noise. It is unclear whether this assumption is upheld by the device in this work:

We thank the Reviewer for raising these important points. In the revised manuscript, we briefly mention it, and added a new section in the revised supplementary Information, which discusses these two important points, that needs to be considered carefully in thermal conductance measurement. In the following paragraph, we will clarify that those requirements indeed were well established in our measurement.

o The width of the floating metal contact appears to be 1 micrometer. What is the traversal time and expected thermalization time for a thin metal film of the deposited type at this temperature?

The dwell time (t_{dwell}) inside a micron-size floating contact was estimated by following the paper by Jezouin, S. et al. Science 342, 601–604 (2013). $t_{dwell} = \frac{D_E \Omega h}{N}$, where D_E the electronic density of states per unit volume per unit energy, Ω is the volume of the micron-size floating contact, and N is the number of channels leaving the floating contact. In our devices, for the typical volume of the floating contact of $\Omega \approx 0.5 \mu\text{m}^3$ and typical density of states for gold $D_E = 10^{47} \text{J}^{-1} \text{m}^{-3}$, we find $t_{dwell} = \frac{40 \mu\text{s}}{N}$. This dwell time is much larger than the typical electron-electron interaction time, which is of the order of ~ 10 ns range for gold at a temperature down to few milli-kelvin, as experimentally demonstrated in F. Pierre et al, Physical Review B 68 085413 (2003). Since the electron-electron interaction time remains orders of magnitude smaller than the dwell time, it firmly establishes the quasi-equilibrium hypothesis that the electrons energy distribution in the micron size ohmic contact is a hot Fermi distribution function characterized by a temperature T_M . We have added a few sentences in the revised manuscript with the above reference, and more details can be found in the revised SI.

o The authors do not appear to have checked whether the reflection coefficient from the floating metal contact is sufficiently small to avoid shot noise contributions to the measured

signal (see i.e. Banerjee, et al., 2017). Why is this the case? Is such data available and consistent with the requirements?

We thank the Reviewer for raising this point. Although we have already shown the data in the Supplementary Information that establishes this fact, we agree that it was not explained sufficiently explicitly. In our device geometry, one can rule out the possibility of significant reflection coefficient by measuring the two charge currents leaving the floating contact towards the reflection and transmission paths. This is shown in the attached figure below. As can be seen, a constant current is injected from the source contact and we measure the voltage drops along the reflected and the transmitted paths, as well as the corresponding resistances ($R_T = \frac{V_T}{I}$ and $R_R = \frac{V_R}{I}$). Both R_T and R_R are identical within the experimental resolution, and are in magnitude exactly half of the respective quantum resistance values of each plateau, which was also shown in the SI – Fig. S6. This guarantees that there is an equal partitioning of the current in our device, which further rules out any detectable reflection from the floating contact. As can be seen in the schematic figure, if there was a reflection because of a bad interface between the graphene and the floating contact, more current will flow along the reflection path compared to transmission one. As a result, one would observe $R_R > R_T$, which was not seen in our experiment. We have highlighted this in the revised manuscript in the “device schematic and response” section and elaborated in the revised SI.

Fig: Left: An illustration of the current distribution in our device after accounting the reflection with coefficient ‘ r ’ from the floating metallic contact. Wiggly line is just shown for the illustrative purpose. It does not represent the microscopic picture of the current reflection. Right: The equipartition of the current on both sides for our device. The red (measured at contact T) and the black (measured at contact R) are exactly falling on top of each other at quantum Hall plateaus. This firmly establishes that the reflection coefficient from the metallic floating contact is at least non-detectable within the experimental resolution. Note that even for 1% reflection would have given $\sim 800\Omega$ resistance difference between the two paths for $1/3$ plateau.

o For both of the above points, can it be ruled out that a change in these properties of the floating contact as a function of bath temperature would lead to the observed behaviour?

As mentioned before, we have ruled out the issues related to the dwell time and finite reflection at the base temperature. With increasing temperature, the dwell time always remains a few orders of magnitude higher than the electron-electron interaction time as the electron-electron interaction time is expected to decrease. Thus, the metallic floating contact remains in the quasi-equilibrium distribution in our experiment. Similarly, the signature of reflection was not seen even at elevated temperatures as can be seen in the figure below, where the equal partitioning holds at 100mK. For completeness, we have added this figure in the revised SI showing the R_{XY} , R_{XX} and partition currents at several temperatures within our working range.

Further, we would like to emphasize that If there would have been any change in the properties of the floating contact as a function of bath temperature, it must affect also the thermal conductance values for the particle-like states (1/3 and 2/5), which is not the case in our experiment. Remarkable agreement of the measured thermal conductance data with the expected theoretical value support that there is no change in the properties of the floating contact as a function of bath temperature.

Fig: Left: Temperature dependence of resistance measured at the source contact S . The quantum Hall plateaus remain the same at 20 mK and 100 mK. Middle: Equipartition of the currents remains the same at 20 mK and 100 mK. Right: The longitudinal resistance $R = V_S/I_R$ also remains the same at 20 mK (black) and 100 mK (red).

- The authors have not presented the temperature dependence of the resistance at the measured filling fractions. (Only two temperature points are shown in a gate voltage sweep in Fig.S5). Is this data available? This would help rule out a change in electrical resistance as the origin of a measured thermal conductance change.

In the previous figure, we presented the data within our working temperature range (20mK and 100mK), and do not observe any detectable changes neither for the Hall nor for the longitudinal resistances. We should like to note that the mobility of our device was $\sim 500000 \text{ cm}^2\text{v}^{-1}\text{s}^{-1}$, for a such a high quality device the typical gaps of the fractional states like 1/3 and 2/3 are of the order of a few kelvins; 2-10 K [see, e.g., Nat Phys 7, 693 (2011), Phys Rev Lett 106, 046801 (2011), Science 337, 1196 (2012), Phys Rev Lett 111, 076802 (2013), Phys Rev Lett 109, 056602 (2012), Phys Rev Lett 121, 226801 (2018), Nat Comm 13 (1), 1-7 (2022)]. Thus, within our working temperature range (<100mK), no resistance change is expected as established in our experiment. As suggested by the Reviewer, we have added some of the

resistance data at intermediate temperatures (60mK) in the revised SI, which is again identical to 20mK and 100mK data.

For this purpose, it would also be helpful to know the specific gate voltage associated with the filling fractions labelled in the figures, which is missing.

We thank the Reviewer for asking this point. The thermal conductance measurement was performed at the middle of the each QH plateau. For completeness, we have included a statement for the same in the revised manuscript.

- No temperature dependence of integer states has been presented. Is such data available? This would be an important check as all the presented data is in the fractional regime.

We thank the Reviewer for asking this. Below we show the thermal conductance data for filling factor 1 at 50 mK, which is very similar to 20mK data as expected for integer QH states. We have added this in the revised SI. This further strengthens the consistency of our data for integer and particle-like fractional QH states, as G_Q is independent of temperature and exhibits no sign of edge reconstruction or counter-propagating modes.

Fig. Black and red solid circles correspond to the data taken at 20 and 50 mK, respectively at integer filling 1. It is clearly evident that the thermal conductance remains the same at 20 and 50 mK bath temperatures. This is expected for an edge structure with only downstream edge modes.

- Why is the bath temperature sweep limited from 20 to 60mK? This seems a narrow window, particularly as the 2/3 state has not attained its asymptotic value, which would be of high interest. If there is not a hard experimental limit at 60mK, can the window be extended?

We thank the Reviewer for asking this question, which is essentially the same as one of questions of Reviewer #2. For convenience, we repeat our reply.

The current experiment involves a metallic floating contact, which is heated to a temperature T_M . As a result, the heat current flows along the ballistic edge channel between this hot

floating metallic contact and cold reservoirs. As mentioned above, the heat leakage from the quantum Hall edges to the environment does not affect the determination of the thermal conductance. At the same time, the heat loss of the floating contact via electron-phonon cooling at an elevated temperature does affect the energy balance on the island: the heat that leaves the island via the edge is then not equal to the dissipated Joule heat. This affects essentially the determination of the thermal conductance value. We have found that for our graphene devices the electron-phonon contribution starts to influence substantially our measurement at floating-contact temperatures (T_M) around 95-100 mK. This can be seen from figure S12 in the SI. Due to this effect, we have restricted our measurement to the bath temperature (T_{bath}) of 60 mK, where the T_M is increased to around 70-80 mK in order to fit a sufficient range of data for the G_Q extraction (Fig. 3 in the main manuscript). One could increase T_{bath} further by 5 to 10 mK, but it will not affect the central result of the current work as for 3/5 we have already seen the thermal conductance plateau for the equilibrated regime, and for 2/3 an asymptotic decay to $\sim 0.5 \kappa_0 T$ for the diffusive nature. Observing the asymptotic value of 2/3 reaching $\approx 0 \kappa_0 T$ will be very interesting at the elevated temperature, but that will be really difficult to achieve for a length of $\sim 5 \mu\text{m}$ channel, as the electron-phonon coupling affects the measurement very significantly at such temperatures. It should be noted that with suitable longer channel length one could achieve this goal. It is also worth noting that even for $\sim 150 \mu\text{m}$ channel length in GaAs, Banerjee, et al., 2017, reported $G_Q \sim 0.33 \kappa_0 T$ for a 2/3 edge at 10mK.

For convenience of the Reviewer, we are including SI figure S12 (below) with this report to emphasize why there is a limitation in thermal conductance measurements at temperature beyond $T_{\text{bath}} > 60\text{mK}$. Note that for our experimental setup, we have $T_{\text{bath}} \approx T_0$ (T_0 is the electron temperature) which is why in the below figure the x-axis is expressed in terms of T_0 .

In response to this question by the Reviewer #3 and a similar question by the Reviewer #2, we have added in the revised version an explanation on the upper border of our temperature range.

Figure: Top panel; left: The thermal current J_Q is plotted as function of $T_M^2 - T_0^2$ for $\nu = 1$ at $T_{\text{bath}} = 20$ mK (corresponding electron temperature $T_0 \sim 23$ mK). The deviation from the linearity is attributed to the electron-phonon cooling. Right: The electronic contribution of the thermal current is subtracted from the total thermal current J_Q which is then plotted as function of $T_M^5 - T_0^5$. Beyond the position of vertical dashed line, which correspond to $T_M \sim 100$ mK of the floating contact, electron-phonon cooling becomes significant.

Bottom panel: Same plots at $T_{\text{bath}} = 50$ mK (corresponding electron temperature $T_0 \sim 52$ mK). Here, also one can see that beyond $T_M \sim 96$ mK of the floating contact, the electron-phonon cooling becomes significant.

REVIEWERS' COMMENTS

Reviewer #1 (Remarks to the Author):

In both the revised manuscript and the reply, the authors have successfully addressed my concerns. Therefore, I recommend a publication of the manuscript in Nature Communications.

Reviewer #2 (Remarks to the Author):

The manuscript has been substantially revised and the authors have provided additional information, such as the data at higher temperatures and more discussions on the edge information. The connection between this work and the previous work from the same group has also been discussed and the authors pointed out that some of the findings were in full agreement. The findings from these two manuscripts can bring out further questions on the relationship between the device size and edge reconstruction. Although I still believe that this manuscript share similarity with the previous publication and they might have been combined together to provide information of edge structure for some FQH states in graphene, I agree that the revised manuscript has made substantial efforts to clarify their findings. I understand that for realistic reasons, experimental results are summarized subsequently and further experimental progress might not be anticipated when the previous manuscript was being prepared.

Reviewer #3 (Remarks to the Author):

I have read the authors' response and changes to the manuscript. They have addressed all questions and I believe the manuscript is now clearer and more convincing. At this point I can recommend publication in Nature Communications.